# Identification of a Novel *SHANK2* Pathogenic Variant in a Patient with a Neurodevelopmental Disorder

**DOI:** 10.3390/genes13040688

**Published:** 2022-04-14

**Authors:** Gabriella Doddato, Alessandra Fabbiani, Valeria Scandurra, Roberto Canitano, Maria Antonietta Mencarelli, Alessandra Renieri, Francesca Ariani

**Affiliations:** 1Medical Genetics, Department of Medical Biotechnologies, University of Siena, 53100 Siena, Italy; gabriella.doddato@dbm.unisi.it (G.D.); alessandra.fabbiani@dbm.unisi.it (A.F.); alessandra.renieri@unisi.it (A.R.); 2Department of Medical Biotechnologies, Med Biotech Hub and Competence Center, University of Siena, 53100 Siena, Italy; 3Genetica Medica, Azienda Ospedaliera Universitaria Senese, 53100 Siena, Italy; mariaantonietta.mencarelli@unisi.it; 4Division of Child and Adolescent Neuropsychiatry, University Hospital of Siena, 53100 Siena, Italy; valeria.scandurra@ao-siena.toscana.it (V.S.); r.canitano@ao-siena.toscana.it (R.C.)

**Keywords:** *SHANK2*, neurodevelopmental disorder, language impairment, exome sequencing (ES)

## Abstract

Genetic defects in the *SHANK2* gene, encoding for synaptic scaffolding protein, are associated with a variety of neurodevelopmental conditions, including autism spectrum disorders and mild to moderate intellectual disability. Until now, limited patient clinical descriptions have been published. Only 13 unrelated patients with *SHANK2* pathogenic variations or microdeletions have been reported worldwide. By Exome Sequencing, we identified a de novo stop-gain variant, c.334C>T, p.(Gln112*), in an Italian patient with a neurodevelopmental disorder. The patient (9 years old) presented the following facial features: a flat profile, thick eyebrows, long eyelashes, a bulbous nasal tip and a prominent columella, retracted ears, dental anomalies. The patient showed speech delay and mild neuromotor delay but not autism spectrum disorder. In conclusion, this patient with a novel pathogenic variant in *SHANK2* enlarges the phenotypic spectrum of *SHANK2*-mutated patients and demonstrates that the severity of *SHANK2*-associated disorders is highly variable.

## 1. Introduction

Neurodevelopmental disorders (NDDs) are a group of heterogeneous conditions affecting 2–5% of children and include autism spectrum disorder (ASD), intellectual disability (ID), developmental delay (DD), and epilepsy [1]. Different phenotypes often coexist in the same patient, thus making classification difficult. The genetic etiology underlying NDDs is highly heterogeneous, with varying degrees of genetic overlap and penetrance or expressivity across the phenotypes [2,3,4,5,6,7,8,9,10].

Genes of the SHANK (SH3 and multiple ankyrin repeat domains protein) family (comprising *SHANK1*, *SHANK2* and *SHANK3*) have been linked to a spectrum of neurodevelopmental disorders. The *SHANK2* gene encodes for a postsynaptic scaffolding protein at glutamatergic synapses in the brain, essential for proper synapse formation, development and plasticity [11,12]. By genome-wide microarray analysis for copy number variants (CNVs) in a German cohort of 184 unrelated individuals with ID and a series of 396 Canadian individuals with ASD, Berkel et al. found one patient in each cohort who had a de novo deletion in the *SHANK2* gene [13]. In the SFARI gene database, *SHANK2* is categorized as a high-confidence autism risk gene. *Shank2* mutant mice exhibit ASD-like behavioral deficits and synaptic dysfunctions [14]. To date, only 13 *SHANK2* mutated patients have been described worldwide, and thus the phenotypic spectrum of the condition is only partially described [15,16,17,18,19,20,21,22,23,24]. Among these patients, all individuals carried de novo variants, six variants resulting in a premature stop codon, and five microdeletions encompassing the gene. All patients had mild to moderate ID, and autism was reported in nearly all of them. There are also reported cases with rare missense *SHANK2* variants inherited from unaffected parents, suggesting the existence of low-penetrance alleles [13].

In the present study, we performed clinical exome sequencing in a patient with a neurodevelopmental disorder and found a de novo *SHANK2* pathogenic variant, c.334C>T, p.(Gln112*). Interestingly, the patient showed mild intellectual disability and speech delay, without any signs of ASD, expanding the phenotypic spectrum associated with *SHANK2* mutations.

## 2. Materials and Methods

### 2.1. Sample and DNA Extraction

Peripheral blood samples were collected in EDTA tubes from the proband and her parents at the Medical Genetics Unit of the Azienda Ospedaliera Universitaria Senese (A.O.U.S, Siena, Italy). Following written informed consent for both diagnostic and research purposes, genomic DNA samples were extracted using MagCore HF16 (Diatech Lab Line, Jesi, Ancona, Italy).

### 2.2. Exome Sequencing

Exome sequencing was performed on genomic DNA samples of the proband and both parents.

Library preparation and exome capture were obtained using the Illumina Nextera Flex for Enrichment KIT according to the manufacture’s protocol, and sequencing was performed using the Illumina NovaSeq 6000 platform (Illumina San Diego, CA, USA). Alignment of raw paired-end reads to the reference genome (version hg19) was performed with BWA Enrichment (v2.1.2). Variant calling was obtained using the Genome Analysis ToolKit (GATK).

### 2.3. Filtering and Variant Prioritization

Exome sequencing data were filtered using eVai software (enGenome v2.3). Variants’ prioritization was obtained by using increasingly enlarged filters: (i) phenotype, using the HPO terms *Delayed speech and language developmental* (HP: 0000750) and *Intellectual Disability* (HP: 0001249); (ii) classification (pathogenic, likely pathogenic and uncertain); (iii) effect (frameshift, stop-gain, splice site and missense variants).

All variants were screened according to their frequency, location, mutation category, literature, and mutation database data (ClinVar database, LOVD database, HGMD database). Polymorphisms (minor allele frequency, MAF < 0.01) were excluded, and synonymous variants were assumed to be benign or likely benign. Missense variants were predicted to be damaging by CADD-Phred prediction tools for functional effect prediction. Frameshift, stop-gain, and splice site variants were prioritized as pathogenic. The following public databases were used for the interpretation of the variants: ClinVar (https://www.ncbi.nlm.nih.gov/clinvar/, accessed on 10 February 2022), LOVD (https://databases.lovd.nl/shared/genes, accessed on 10 February 2022), the Human Genome Mutation Database (HGMD, http://www.hgmd.cf.ac.uk/ac/index.php3, accessed on 10 February 2022), Varsome (https://varsome.com/ accessed on 10 February 2022).

Finally, for the interpretation of the variants, the American College of Medical Genetics and Genomics (ACMG) 2015 guidelines were used [25].

### 2.4. Sanger Sequencing

To confirm the *SHANK2* variant, we performed Sanger sequencing of exon 3, including the flanking intron sequences of the gene (NM_133266.3) in the proband and her parents. DNA was amplified by PCR using specific primer pairs. Subsequently, the purified PCR products were applied to Sanger sequencing to affirm the mutation, using an ABI PRISM3130 Genetic Analyzer (Applied Biosystems, Foster City, CA, USA). The Sanger sequencing results were analyzed with Sequencher software V.4.9 (Gene Codes, Ann Arbor, MI, USA).

### 2.5. Mutation Nomenclature

The mutation is described according to Human Genome Variations Society (HGVS). Nucleotide numbers are derived from the cDNA sequence of *SHANK2* (GenBank accession NM_133266.3).

## 3. Results

### 3.1. Clinical Description

The patient was a 9-year-old female, the first child of healthy parents.

She was born from an uncomplicated pregnancy at full term. Family history was negative for ASD or other neurodevelopmental disorders. The auxological parameters at birth were: length 51 cm (71° percentile) and weight 3450 g (25°–50° percentile).

The mother reported that the baby showed hypovalid suction. She described the onset of crawling at 12 months, autonomous walking at 15 months, production of single words at 2 years and 6 months and production of simple sentences at 6 years. Sphincter control was acquired at about 3 years. Eye contact with parents has always been good. At the age of about 2 years and 6 months, the girl began psychomotor therapy and speech therapy, with significant benefits. The girl has support at school and shows considerable difficulties in mathematical calculation. Good interaction with peers is reported. Her diet is varied and her sleep–wake rhythm is normal. She underwent audiometric, ophthalmological, brain MRI and EEG evaluation, all of which were normal.

Physical examination showed height of 127 cm (10° percentile), weight of 24 kg (5°–10° percentile) and head circumference of 51 cm (10°–25° percentile). Facial dysmorphisms included: a flat profile, thick eyebrows, long eyelashes, a nose with a bulbous tip and a prominent columella, large and spaced teeth, retracted ears (Figure 1).

### 3.2. Genetic Analysis

Exome Sequencing analysis was performed on the patient and her parents. We obtained a mean depth of coverage of 80X. A total of 11.670 genetic variants on average for each sample was yielded.

The analysis by classification filtering revealed a de novo stop-gain variant c.334C>T, p.(Gln112*) in exon 3 of the *SHANK2* gene on chromosome 11 (Figure 2).

Sanger sequencing confirmed the heterozygous variant in the proband and not in parents (Figure 3). This frameshift deletion has not been previously reported and it is absent in the ExAC and gnomAD databases. The variant has been submitted to the LOVD database (https://www.LOVD.nl, accessed on 10 February 2022) with the following ID: 0000836786.

## 4. Discussion

In the present study, we report the identification of one rare patient with a *SHANK2* pathogenic variant, c.334C>T, p.(Gln112*), showing a neurodevelopmental disorder. To date, only 13 cases with de novo *SHANK2* alterations have been reported worldwide, and the resulting phenotypes are poorly described (Table 1) [15,16,17,18,19,20,21,22,23,24].

All reported patients had ID, ranging from mild to moderate [15,16,17,18,19,20,21,22,23,24]. In accordance, the girl described here showed mild intellectual disability. Nearly all previous patients showed autism, while our case does not show any sign of ASD. Our patient manifested language delay, and this represents a key feature of the condition being present in all reported patients [15,16,17,18,19,20,21,22,23,24]. Additional neurologic features such as anxiety or repetitive behavior were frequently observed but were absent in the present case [13,15,16,21,22,23,24]. The delineation of a common facial phenotype is hampered by the lack of facial descriptions in previous papers. Here, the patient shows a flat profile, thick eyebrows, long eyelashes, a nose with a bulbous tip and a prominent columella, large and spaced teeth, retracted ears.

This variant, c.334C>T, p.(Gln112*), in *SHANK2* is an early truncating variant with abolishment of the major domains of the protein (SH3, PDZ domain, Proline-rich region, SAM). In accordance with the present case, *SHANK2* truncating variants are all de novo events in affected individuals. Only *SHANK2* missense variants found in patients with NDD are inherited from healthy parents and probably represent low-penetrance alleles [13]. Dhaliwal et al. identified a missense variant in the *SHANK2* gene in association with multiple inherited variants in other ASD-associated genes (*RELN*, *SHANK2*, *DLG1*, *SCN10A*, *KMT2C* and *ASH1L*) in a family with three affected children and hypothesized that additional genetic/epigenetic factors together might be necessary to develop ASD [26]. Moreover recent studies support that genetic defects in the *SHANK2* gene are implicated in neuropsychiatric disorders [27,28,29]. The mutation type as well as the presence of modifiers in the genome can be fundamental. Leblond and colleagues recently identified CNVs as putative modifiers of the phenotype in accordance with a “multiple-hit model” [18]. Additional studies are necessary to delineate more robust genotype–phenotype correlations.

In *Shank2* mutant mouse models, anxiety-like behavior, hyperactivity, abnormal social behavior and dysregulation of synaptic molecules have been observed in specific regions of the brain [30].

A possible pathogenic mechanism has been elucidated by the use of human induced pluripotent stem cells (hiPSCs). Lutz et al. generated hiPSCs from a patient carrying a heterozygous deletion of *SHANK2* and from the unaffected parents. They observed a reduction in growth cone size and a transient increase in neuronal soma size during neuronal maturation in patients’ derived cells. Furthermore, the extracellular signal-regulated kinase (ERK) pathway resulted dysregulated [31]. Another hiPSC study showed that human neurons with ASD-associated haploinsufficient *SHANK2* mutations had increased synaptic connectivity relative to controls [32]. These findings reinforce the hypothesis of *SHANK2* involvement in neurodevelopmental disorders. In addition, the generation of animal models and hiPSC studies are important to discover and design therapeutic strategies.

Regarding the treatment of ASD, the supplementation of zinc (Zn) is a possible preventive strategy. Zinc is a metal involved in the developmental and functioning of the central nervous system (CNS) and is crucial for normal cognitive functions and emotional behaviors. Zn is also responsible for maintaining viable synapses by stabilizing SHANK2/3 postsynaptic scaffolding proteins [33,34]. Zn deficiency is a risk factor that may contribute to the pathophysiology of neurodevelopmental disorders such as ASD. Zinc supplements improved the behavioral deficits in animal models of ASD and in Shank3-mutated animals [35]. Clinical trials are still needed to validate the beneficial therapeutic effects of zinc supplements in ASD patients.

In conclusion, our findings underline that ASD is not necessarily a constant clinical feature in *SHANK2*-mutated patients and, in accordance with previous literature, indicate language impairment as a major feature in these patients. A peculiar facial phenotype may be observed, but additional studies with comprehensive clinical descriptions are required.

## Figures and Tables

**Figure 1 genes-13-00688-f001:**
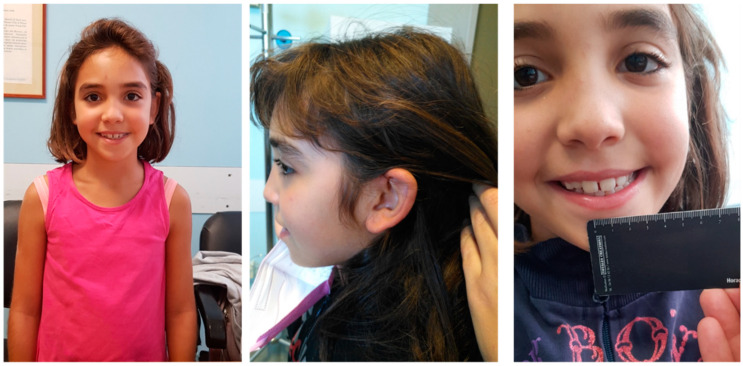
Photographs of the proband showing the main clinical features: thick eyebrows, long eyelashes, a nose with a bulbous tip and a prominent columella, flat profile, retracted ears, large and spaced teeth.

**Figure 2 genes-13-00688-f002:**
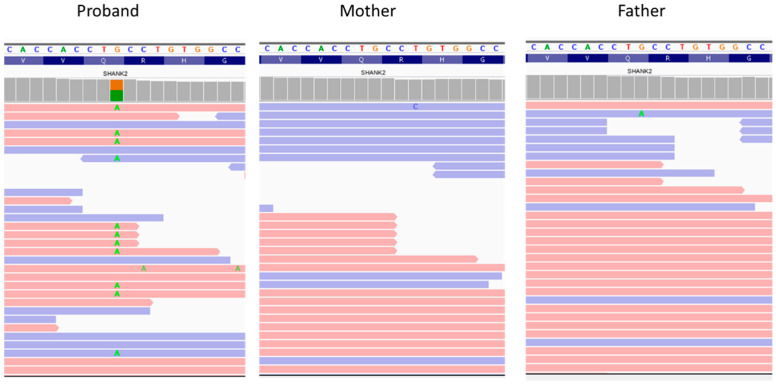
Visualization of the variant in *SHANK2* is show with an integrative genomics viewer. The variant c.334C>T was heterozygous in the proband and absent in the parents.

**Figure 3 genes-13-00688-f003:**
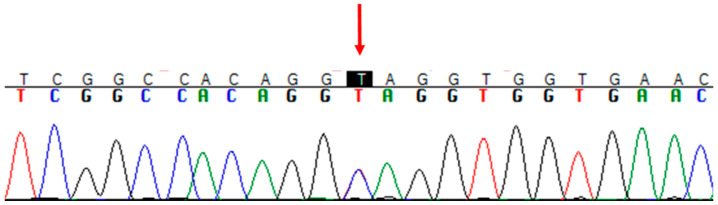
Confirmation of the variant c.334C>T, conducted by Sanger sequencing.

**Table 1 genes-13-00688-t001:** Clinical features of the present case compared with the previously reported patients with *SHANK2* variants.

	Present Study	Caumes et al., 2020		Zhou et al., 2019	Guo et al., 2018	Bowling et al., 2017	Marcou et al., 2016	Leblond et al., 2014	Leblond et al., 2012	Wischmeijer et al., 2011	Pinto et al., 2010	Berkel et al., 2010		
	**Pat.1**	**Pat.1**	**Pat.2**	**Pat.1**	**Pat.1**	**Pat.1**	**Pat.1**	**Pat.1**	**Pat.1**	**Pat.1**	**Pat.1**	**Pat.1**	**Pat.2**	**Pat.3**
**AGE**	9	6	6	4	NA	NA	12	NA	11	8	NA	NA	NA	NA
**GENDER**	F	M	F	M	M	NA	F	M	M	F	M	F	M	M
**VARIANT**	c.334C>T	c.1322del	c.132dup	c.2540_2541del	c.87C>G	c.1896dup	11q13.2 to 11q13.4	del_11q13.3q13.4 (all *SHANK2* exons)	loss of exon 5-16	del_11q13.3q13.4 (all *SHANK2* exons)	del exon 5-16	del exon 7	del exon 7-6	c.2521C>T
	p.(Gln112*)	p.(Ile441Thsfs*8)	p.(Asp45Argfs*3)	p.(Ser847*)	p.(Tyr29*)	p.(Asp633Argfs*3)	p.(?)	p.(?)	p.(?)	p.(?)	p.(?)	p.(?)	p.(?)	p.(Arg841*)
**PATTERNS OF INHERITANCE**	de novo	de novo	de novo	de novo	de novo	de novo	de novo	de novo	de novo	de novo	de novo	de novo	de novo	de novo
**SPEECH DELAY**	+	+	+	+	+	+	+	+	+	+	+	+	+	NA
**AUTISM/ASD**	−	ASD	AUTISM	ASD	ASD	NA	NA	AUTISM	AUTISM	AUTISM	AUTISM	AUTISM	AUTISM	AUTISM
**ID**	−	Mild	Moderate	NA	+	Moderate	Moderate/ Severe	Severe	Moderate	Moderate/Severe	Mild	Moderate	Mild	Moderate
**MOTOR DELAY**	−	−	−	−	−	NA	+	+	+	+	NA	+	NA	NA
**NEUROLOGICAL SIGNS**	Difficulties in mathematical calculation	Sleep and attention disorders	Anxiety	Sleep disorder	Attentional problems Febrile seizures infancy repetitive behavior	Hyperactivity	−	Oral dyspraxia Signs of cerebellar dysfunction Slight hypotonia	−		NA	Slow reaction and limited mimicry	NA	NA
**CLINICAL EXAMINATION**	Flat profile, thick eyebrows, long eyelashes, bulbous tip and prominent columella, large and spaced teeth, retracted ears	NA	NA	NA	NA		Microcephalic Mild malar hypoplasia, mild retrognathia. Fine hair. Prominent forehead Bilateral epicanthal folds, long palpebral fissures, deep-set eyes. Broad nasal tip, depressed nasal bridge. Small mouth, down-turned corners. Wide-spaced teeth. Long slender fingers, clinodactyly. Long, slender feet, mild pes planus, piezogenic papules	Clinodactyly Deep-set eyes strabismus and ptosis. Large ears Retrognathia Wide nasal bridge Thin upper lip	Prominent chin, hypermetropia and astigmatism	Congenital hip dysplasia, downward slanting palpebral fissures, deep-set eyes, ptosis of the left eyelid, long and fine lashes, broad nasal bridge, tubular nose with round overhanging tip and hypoplastic nares, short philtrum, small and thin upper lip, preauricular tag and small low-set simple ears	Hypermetropia, large and prominent ears, flat feet	NA	NA	NA
**MRI**	Normal	Normal	Normal	NA	−	NA	Normal	NA	NA	Normal	Normal	NA	NA	NA
**EEG**	Normal	NA	NA	−	−	NA	Normal	NA	Normal	Normal	NA	NA	NA	NA

* means change in a stop codon; NA, not available; M, male; F, female; +, present; −, absent.

## Data Availability

The data presented in this study are available on request from the corresponding author.

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
