# Peer review of "Identification of a Novel SHANK2 Pathogenic Variant in a Patient with a Neurodevelopmental Disorder"

_genes, 2022, doi:10.3390/genes13040688_

Round 1
Reviewer 1 Report
The article reads well. The data are cleary stated. Table 1 is welcome so that the clinical features of the patient can be compared to the clinical features of all other published patients.
The presentation of this case report is fine.
Author Response
Dear Reviewer 1,
thanks for your positive opinion.

Reviewer 2 Report
The authors described a single patient with mild neurological symptoms and identified a new de novo truncating mutation in SHANK2 gene. Although the identification of a new SKANK2 mutation could be useful for clinicians and medical geneticists, the MS is truly essential and it does not cited the last papers in the field of SHANK2. The authors should improve their case report by discussing the most recent findings reported in the literature. Alternatively, the authors could review the literature and include their patient's description. Therefore, they may present a more recent literature review and not be limited to just a description of a new clinical case.
1.0.0.20Author Response
Dear Reviewer 2,
thank you for your suggestions.
We have added the most recent findings reported in the literature about animal models and hiPSC study (Line 45-46 and line 174-186; Lee et al., Mol Brain. 2020; Lutz et al., Front Mol Neurosci. 2021; Zaslavsky et al. Nat Neurosci. 2019).

Round 2
Reviewer 2 Report
Many thanks. I appreciate the effort of the authors but in my opinion the MS requires an extensive revision.
1.0.0.20 1.0.0.20